# Work-related psychosocial risk factors and psychiatric disorders: A cross-sectional study in the French working population

**Astrid M. Chevance**[1]*, **Oumou S. Daouda**[2], **Alexandre Salvador**[3,4], **Patrick Légeron**[3], **Yannick Morvan**[3,4], **Gilbert Saporta**[5], **Mounia N. Hocine**[2☯], **Raphaël Gaillard**[3,4,6☯]

**1** University of Paris, CRESS, INSERM, INRA, Paris, France, **2** Laboratoire—Modélisation Epidémiologie et Surveillance des Risques Sanitaires (MESuRS), CNAM Conservatoire National des Arts et Métiers, Paris, France, **3** Service Hospitalo-Universitaire de Psychiatrie, Centre Hospitalier Sainte-Anne, Paris, France, **4** Inserm U894, Centre de Psychiatrie et Neurosciences, Paris, France, **5** Cédric (Centre d'Études et de Recherches en Informatique et Communication), Paris, France, **6** University of Paris, Paris, France

☯ These authors contributed equally to this work.
* astrid.chevance@gmail.com

**Data Availability Statement:** All relevant data are within the manuscript and its Supporting Information files.

## Abstract

### Purpose

The study estimates the prevalence of probable psychiatric disorder in the working population, determines the proportion of people presenting a probable psychiatric disorder among people exposed to work-related psychosocial risk factors (PSRFs), and identifies which PSRF has the strongest association with having a probable psychiatric disorder.

### Methods

A cross-sectional study conducted in March 2018 involved a representative sample of the French working population. The General Health Questionnaire 28 (GHQ-28) was used to estimate the prevalence of probable psychiatric disorder and 44 items were gathered from theoretical models of PSRFs. We used multiple logistic regression to estimate the association of each PSRF with having a probable psychiatric disorder, adjusted on individual, health, and job confounders.

### Results

This study involved 3200 French participants. The proportion of probable psychiatric disorder was 22.2% [20.6; 24.0]. Ten PSRFs were significantly associated with it. The strongest association was for having problems handling professional and personal responsibilities (reported by 15% of the study population) (OR = 1.97 [1.52; 2.54]), with 45% pathological GHQ-28 scores (potential psychiatric cases) for people exposed to this PSRF versus 18% non-exposed. The next strongest association was lack of support of colleagues (reported by 28%) (OR = 1.63 [1.29; 2.06]). The third strongest association was feeling sometimes afraid when doing the job (reported by 63%) (OR = 1.53, [1.21; 1.93]).

**Funding:** This work was supported by the Fondation Pierre Deniker, a non-profit French foundation for mental health (65742000). https://www.fondationpierredeniker.org/ Yes the funder has a role in the decision to publish.

**Competing interests:** "None": Oumou Daouda, Mounia N. Hocine, Yannick Morvan and Alexandre Salvador report no conflict of interest Astrid Chevance received a PhD grant from La Fondation pour la Recherche Médicale (FMD20170637634) Raphaël Gaillard has been a member of a scientific board for Janssen, Lundbeck, Roche, SOBI and Takeda. He has been a consultant or speaker for Astra Zeneca, Boehringer-Ingelheim, Pierre Fabre, Lilly, LVMH, Lundbeck, MAPREG, Otsuka, Pileje, Sanofi, Servier and has received professional fees and has received funding for research from Servier. He is a founding member of Regstem. He is the president of the Fondation Pierre Deniker. Gilbert Saporta is currently a consultant for Ipsos. Patrick Légeron is a founding member of Stimulus. This does not alter our adherence to PLOS ONE policies on sharing data and materials.

## Conclusions

Our study identified 10 PSRFs associated with psychiatric disorder, with substantial exposure rate among the population. The results of our research could help develop recommendations to improve work environment.

## Introduction

Treatable common psychiatric disorders affected 17.3% of people in European Union in 2018. Anxiety disorders (5.4%) and depressive disorders (4.5%) were the most prevalent [1]. Psychiatric disorders are also an important part of the global burden of disease: for instance, major depressive disorder was one of the five leading causes of years lived with disabilities in 2016, contributing to 4.2% of the global disease burden [2].

Psychiatric disorders seem to have multiple causes, and the impact of work-related psychosocial risk factors (PSRFs) on mental health outcomes have been investigated for 40 years. For example, conceptual models such as the job demand-control-support model, the effort-reward imbalance model and the organizational justice model theorized the link between job determinants and mental health outcomes [3–5]. Epidemiological studies have investigated the association between PSRFs and a range of common mental disorders such as generalized anxiety disorder or alcohol abuse [6,7]. A recent meta-analysis showed that being exposed to PSRFs such as "poor supervisor and colleague support" or "job insecurity" was correlated with suicidality outcomes [8].

However, when the association between psychiatric disorders and work-related PSRF was investigated, it was mostly for specific disorders and with a specific population, mainly sick people [9]. Studies of representative samples of the working population are required for at least two reasons: they allow for a robust generalization of results as well as robust analyses of the association between work-related PSRFs and mental health disorders because they include not just sick individuals.

This STROBE-compliant cross-sectional study of a representative sample of the French workers aimed to estimate the prevalence of probable psychiatric disorders in the working population, determine the proportion of people presenting a probable psychiatric disorder among people exposed to work-related PSRFs, and determine which PSRF has the strongest association with having a probable psychiatric disorder, after adjusting for individual health and job characteristics.

## Material and methods

The report of the study is compliant with the STROBE checklist for cross-sectional studies [10].

### Design and population

A cross-sectional study was performed between February 27 and March 6, 2018 with 3200 individuals living in France and representative of French workers. Individuals aged 18 to 80 years who declared currently having a job (even a part-time job) whatever their occupation or status (employee or self-employed) were eligible. We excluded students, unemployed individuals, housewives/husbands and retired people. People were recruited in the "Ipsos Access Panel" and were invited by e-mail to participate in the study. Representativeness was achieved

by using quota sampling for sex, age, occupation and residence locality and the rim weighting method (raking) [11]. All data were collected by using a standardized questionnaire (S1 Table) administered during a computer-assisted web interview.

## Ethics

The "IPSOS Access Panel" is acknowledged by the French National Institute for Data Security (CNIL) and is compliant to the ESOMAR international code of conduct. All participants answered the questionnaire voluntarily after giving their informed consent online and could unsubscribe from the survey without influencing panel membership. Since the study did not involve any human experimentation and the online questionnaire was proposed to volunteers of the IPSOS Access Panel, in respect with data protection standards of the French law, we did not seek for the approval of an ethic committee.

## Data collection

**Measurement of probable psychiatric disorder.** We used the General Health Questionnaire– 28 (GHQ-28), a self-administered screening questionnaire designed to detect a probable psychiatric disorder in primary care settings [12]. The GHQ-28 is meant to estimate prevalence in a large population more than for individual cases. The questionnaire explores four dimensions—depressive symptoms, anxiety symptoms, social dysfunction and psychosomatic symptoms—with a list of 28 items, each one rated "better than usual", "same as usual", "worse than usual" and "much more than usual". The Likert scoring is recommended (scoring each item from 0 to 3 with a global score ranging from 0 to 84) [13]. A total score of 24 is the threshold for detecting probable psychiatric disorder with sensitivity 79.8% and specificity 78.5% [14]. The French version of the questionnaire, used in our study, has been validated [15].

**Measurement of psychosocial risk factors (PSRF).** PSRF were evaluated with a list of 44 questions (S1 Table) were compiled by the scientific committee according to major theoretical works from Karasek, Siegrist and Greenberg. In 1979, Karasek showed that a high job demand associated with low job control (decision latitude) is associated with high mental strain as well as medical outcomes [4]. This model was enriched in 1990 with social support that appeared to be stress-buffering [5]. In 1996, Siegrist added the effort-reward imbalance model to assess adverse health effects of stressful experience at work [16]. Another complementary theoretical framework is the model of organizational justice that assess fairness perceptions of organizations [3,17]. All these models describe the individual perception of the work environment. Moreover, we took into account other factors identified in the literature such as organizational change, job insecurity, workplace conflict and bullying [18]. The 44 items were rated "utterly true", "quite true", "quite false", "utterly false". The scoring method was to 1 for "utterly true" and "quite true" and 0 for "quite false" and "utterly false". Therefore, participants with a score of 1 to a given PSRF are called "participants exposed" to this PSRF in the results section. Among the 44 questions, 35 concerned only employees and not self-employed individuals.

**Individual and job characteristics.** Several groups of variables about individual factors were measured to describe the population:

- **Socio-demographic factors (8 variables)**: age, sex, residency, marital status, number of children, highest educational degree obtained, household annual income, occupational status

- **Health and life hygiene conditions (6 variables)**: chronic disease (>6 months), psychotropic medication in the past 12 months, medical work cessation over 1 week due to a

psychiatric condition in the last 12 months, alcohol consumption, smoking, illegal drug use in the past 12 months.

- **Job characteristics (10 variables)**: activity sector, employed or self-employed status, duration of the job (months), work duration (hours per week), weekend work, night work, staggered hours, commuting duration, teleworking, previous experience with unemployment.

- **Work environment (2 variables)**: workplace (e.g., open space, co-working, at home, etc.), company/public institution size.

## Statistical analysis

The study outcome was the binary variable of having or not a probable psychiatric disorder defined by the individual GHQ-28 score $\geq$ 24 [14]. We assessed the prevalence of probable psychiatric disorder: the distribution of the GHQ-28 global score, mean and median for the sample were calculated and compared to the cut-off $\geq$ 24. We also assessed exposure to psychosocial risk factors and prevalence of probable psychiatric disorder: the descriptive statistics for all variables are given as percentages for categorical variables or mean (SD) for continuous variables.

For each PSRF, we calculated the proportion of people exposed and not exposed to the PSRF, and for each category (exposed/not exposed), the prevalence of probable psychiatric disorder. The differences in prevalence between exposed and non-exposed groups were tested by chi-square test. $P<0.05$ was considered statistically significant with Holm-Bonferroni correction.

For individual and job-related variables, we calculated the proportion of people with GHQ-28 score $\geq$24 (potential psychiatric cases) and tested the difference in proportion of probable psychiatric disorder between exposed and non-exposed groups for each variable by chi-square test. $P<0.05$ was considered statistically significant with Holm-Bonferroni correction. For significant variables with more than two categories, a Pearson residuals table [(observed–expected)/sqrt (expected)] was computed to identify the modality with a significantly higher proportion of GHQ-28 score $\geq$ 24. This modality is the one with the higher Pearson residual for GHQ score $\geq$ 24.

To investigate the association between each PSRF and probable psychiatric disorder, we modeled the binary outcome GHQ-28 score $\geq$24 as a logistic regression of some PSRFs, adjusted for possible confounders found in the literature: age, sex, chronic medical condition, working on the weekend, staggered work hours, working at night, previous experience with unemployment, work duration per week, and commuting duration [18,19]. To select which PSRF belonged to the model, we used a fast-backward procedure [20]. This method is extremely efficient because the model is not refitted for every variable removed. The model was used for employees, excluding self-employed individuals because the latter group had 8 PSRFs less measured.

Because sex was previously identified as a confounding factor for psychiatric disorder, we calculated the interactions between sex and the 11 PSRFs selected by the fast-backward procedure and included in the model [19]. We also calculated the 55 interactions between the 11 PSRFs included in the model by using the procedure proposed by Cox and Wermuth [21]. This procedure gives an idea of the overall importance of the interaction term in the model by estimating and testing all interaction terms desired, one at a time. We corrected the p value with Holm-Bonferroni.

## Results

### Description of the sample and prevalence of psychiatric disorders

Among the 40514 eligible people from the "IPSOS Access Panel" invited by e-mail to participate in the study, 10660 agreed and 3487 were included (S1 Fig). Because 287 did not complete

**Table 1. Characteristics of the whole sample and of participants with a GHQ-28* score ≥ 24.**

| | Whole sample (n = 3200) | GHQ28 score ≥24 (n = 688) |
|---|---|---|
| **Sex** | | |
| Male | 51.6 | 18.8 |
| Female | 48.2 | 25.8** |
| **Age, years** | 41.4 (11.1) | 39.8 (10.9) |
| **Number of children** | | |
| None | 52.6 | 21.5 |
| 1 | 21.7 | 23.7 |
| 2 | 19.9 | 23.9 |
| 3 | 4.8 | 27.2 |
| > 4 | 0.9 | 15.3 |
| **Marital status** | | |
| Married/free union | 71.5 | 21.2 |
| Alone | 28.5 | 24.8 |
| **Highest educational degree** | | |
| < baccalaureate degree | 21.3 | 20.3 |
| Baccalaureate degree | 21.7 | 20.1 |
| Baccalaureate degree +2 years | 24.2 | 23.6 |
| Baccalaureate degree+ ≥ 3 years | 32.8 | 23.8 |
| **Annual household income (euros)** | | |
| < 15 000 | 11.2 | 30.9** |
| 15–24 000 | 20.5 | 24.2 |
| 24–36 000 | 26.2 | 22.5 |
| > 36 000 | 29.3 | 19.3 |
| No information | 12.8 | 17.4 |
| **Occupational status (independent worker)** | N = 400 | |
| Farmer | 18.3 | 15.9 |
| Craftsman, shopkeeper, entrepreneur | 47.5 | 21 |
| Managerial and professional occupation | 18.3 | 17.8 |
| Technician and associate professional | 15.9 | 20.3 |
| **Occupational status (employees)** | N = 2800 | |
| Managerial and professional occupation | 17.2 | 20.6 |
| Technician and associate professional | 27.7 | 24.6 |
| Service and office worker | 31.5 | 22.8 |
| Worker | 23.6 | 21.5 |
| **Self employed** | | |
| Yes | 11.2 | 19.9 |
| No | 88.8 | 22.5 |
| **Size of company (no. of employees)** | | |
| < 10 | 22 | 21.9 |
| 10–49 | 22.6 | 23 |
| 50–99 | 10.6 | 25.3 |
| 100–499 | 20.4 | 21.1 |
| > 499 | 16.4 | 22.3 |
| No information | 7.9 | 19.6 |
| **Activity sector** | | |
| Industry | 12.7 | 21.3 |

(*Continued*)

**Table 1.** (Continued)

|  | Whole sample (n = 3200) | GHQ28 score ≥24 (n = 688) |
|---|---|---|
| Building | 11.2 | 20.2 |
| Trading | 11.8 | 20.9 |
| Transport | 5.6 | 26.3 |
| Insurance and real estate | 3.4 | 28 |
| Education, health and social work | 23.5 | 23.9 |
| Other services | 31.8 | 21.2 |
| **Duration of job** |  |  |
| < 6 months | 5.6 | 19.7 |
| 6 months–5 years | 33.6 | 24.1 |
| 6–10 years | 21.7 | 21.3 |
| > 10 years | 39.1 | 21.4 |
| **Workplace** |  |  |
| Individual office | 23.1 | 19.8 |
| Open space | 17.5 | 24.8 |
| Flex office | 2.4 | 33.8 |
| At home | 5.9 | 22.3 |
| Production or workshop | 9.1 | 22.1 |
| External sites | 8.3 | 16.8 |
| At customer | 7 | 26.9 |
| Commercial permises | 8.1 | 22.6 |
| Other | 18.6 | 21.7 |
| **Teleworking** |  |  |
| Yes | 12.4 | 25 |
| No | 81.6 | 21.8 |
| **Work duration (hours per week)** |  |  |
| < 39 | 57.4 | 22.5 |
| 39–50 | 37.5 | 20.1 |
| > 50 | 5.1 | 34.4** |
| **Working on the week-end** |  |  |
| Yes | 64.2 | 23.1 |
| No | 35.8 | 20.5 |
| **Night work** |  |  |
| Yes | 28.5 | 25.3 |
| No | 71.5 | 21 |
| **Staggered hours** |  |  |
| Yes | 61.4 | 23.9 |
| No | 38.6 | 19.4 |
| **Commuting duration** |  |  |
| < 30 min | 59 | 19.6 |
| 31 min–1hr | 21.9 | 24.1 |
| > 1hr | 19.1 | 27.9** |
| **Previous experience with unemployment** |  |  |
| Never | 51 | 20.3 |
| 1 episode | 20.8 | 21.5 |
| ≥ 2 episodes | 28.2 | 26.2** |

(*Continued*)

**Table 1.** (Continued)

|  | Whole sample (n = 3200) | GHQ28 score ≥24 (n = 688) |
|---|---|---|
| **Sickness absence over 1 week due to psychiatric conditions in the last 12 months** |  |  |
| Yes | 12.7 | 50.3** |
| No | 87.3 | 18.1 |
| **Chronic disease (> 6 months)** |  |  |
| Yes | 23.9 | 36.8** |
| No | 76.1 | 17.6 |
| **Psychotropic medication in the past 12 months** |  |  |
| Yes | 17.2 | 47.7** |
| No | 82.8 | 16.9 |
| **Alcohol consumption per week** |  |  |
| > 10 glasses | 9.2 | 25.8 |
| > 21 glasses | 3.1 | 21.5 |
| **Illegal drug use in the past 12 months** |  |  |
| Yes | 6.3 | 44.4** |
| No | 92.9 | 20.5 |
| **Smoking** |  |  |
| Yes | 24.9 | 24.8 |
| No | 75.1 | 21.3 |

Data are proportions (%) for categorical variables and mean ± SD for continuous variables. Differences in proportions were tested by chi-square test. For significant variables with more than two categories, a Pearson residuals Table [(observed–expected)/sqrt(expected)] was computed to identify the modality with a significantly higher proportion for GHQ-28 score ≥ 24. This modality had the higher Pearson residual for GHQ score ≥ 24.

*GHQ-28 = General Health Questionnaire– 28

** indicates significance according to Holm-Bonferroni correction.

the questionnaire to the end, data for 3200 people were analyzed. For the following analysis, the sample was weighted to be representative of the general population (S2 Table).

For the whole sample (Table 1), the mean (SD) mean age was 41.4 (11.1) and 48.2% were women. Overall, 42.6% of participants had a work duration of more than 39 hours a week; in France, the legal work duration is 35 hours a week. Also, 28.5% declared night work, 64.2% weekend work and 61.4% staggered hours; 19.1% reported more than 1 hour commute time to work. In all, 51% reported no previous experience with unemployment, whereas 28.2% had ≥ 2 experiences. Also, 12.7%, had a sickness work cessation over 1 week in the last 12 months due to a psychiatric condition. Moreover, 17.2% reported having taken psychotropic medication (e.g., antidepressants, anxiolytics) and 6.3% reported using illegal drugs in the past 12 months. Furthermore, 23.9% reported a chronic disease (physical or mental). Finally, 9.2% reported drinking > 10 glasses per week, which is a cut-off for high risk of all-cause-mortality recently identified [22]. In addition, 3.1% declared drinking > 21 glasses of alcohol per week, which is the classical cut-off for the WHO.

The distribution of scores for the GHQ28 is represented in Fig 1. The mean (SD) GHQ-28 score for the whole sample was 18.3 (13.9) and the median score was 15 [14]. With the threshold score of 24 recommended in the literature the prevalence of probable psychiatric disorder was 22.2% (95% CI [20.6; 24.0]) [14]. The proportion of probable psychiatric disorder was increased for women (25.8%) and people with an annual household income < 15 000 euros

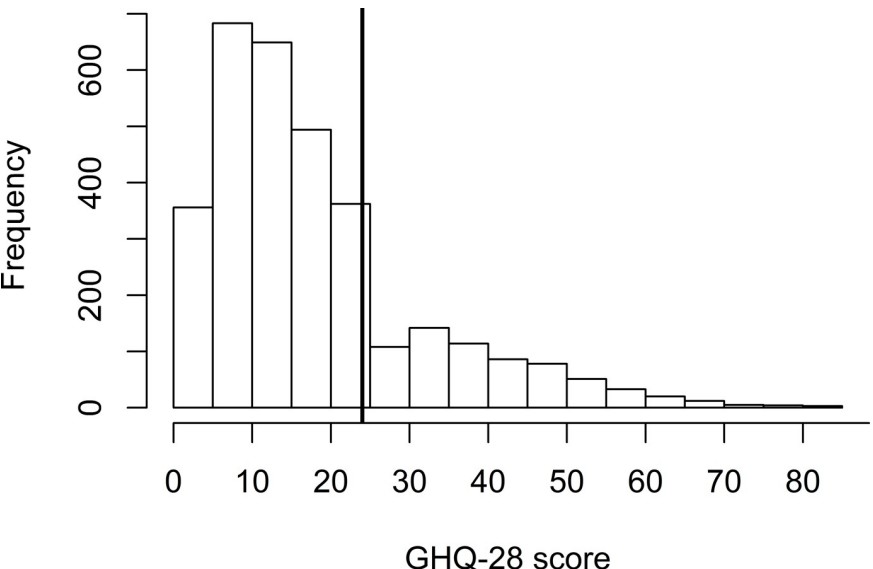

**Fig 1. Distribution of the GHQ-28 score among the whole sample (n = 3200).** The vertical line represent the cut-off 24 we chose according to literature to identify probable psychiatric disorders (GHQ-28 score ≥24).

(30.9%) as well as among people with a chronic disease for more than 6 months (36.8%). This proportion was also increased for people working > 50 hours a week (34.4%), those with a commuting duration > 1 hour (27.9%) and those with ≥ 2 previous experiences with unemployment (26.2%).

## Exposure to work-related PSRFs

Fig 2 presents the ranking of the PSRFs according to the absolute difference between the proportion of probable psychiatric disorder among individuals exposed and non-exposed to the PSRFs as well as the proportion of exposure to PSRFs for the whole sample. For all PSRFs, the difference in proportion of probable psychiatric disorder between exposed and non-exposed individuals was significant, except for "The job I do requires that I constantly adapt to new things" and "My job often puts me in contact with clients/users". These two PSRFs were the most reported, with 75.7% of the sample reporting having a job that required constantly adapting to new things and 70.5% reporting often being in contact with customers/clients/users.

In contrast, "I have problems handling personal and professional responsibilities" was the least reported PSRF (15.4% of the whole sample). However, for this PSRF, the absolute difference between the prevalence of probable psychiatric disorder among exposed (44.6%) and non-exposed (18.1%) individuals was the largest among the 44 PSRFs. The second largest absolute difference in proportion of cases between exposed and non-exposed individuals (21.3 points) was for "My work environment is unpleasant" (22.4% of the sample) and "sometimes I feel afraid when I do my job" (27.8% of the sample). The proportion of probable psychiatric disorder among people reporting an unpleasant work environment was 38.7% and only 17.4% among those who reported a pleasant environment. The proportion of probable psychiatric disorder among people who declared feeling sometimes afraid when they did their job was 37.6% versus 16.3% for those not exposed. The third largest absolute difference (20.3 points) between the proportion of probable psychiatric disorder among exposed and non-exposed individuals was "My job does not makes me feel useful nor gives me self-esteem" (20.7% of the sample) and "I don't get well along with my hierarchy" (18.7% of the sample). In all, 38.3% of

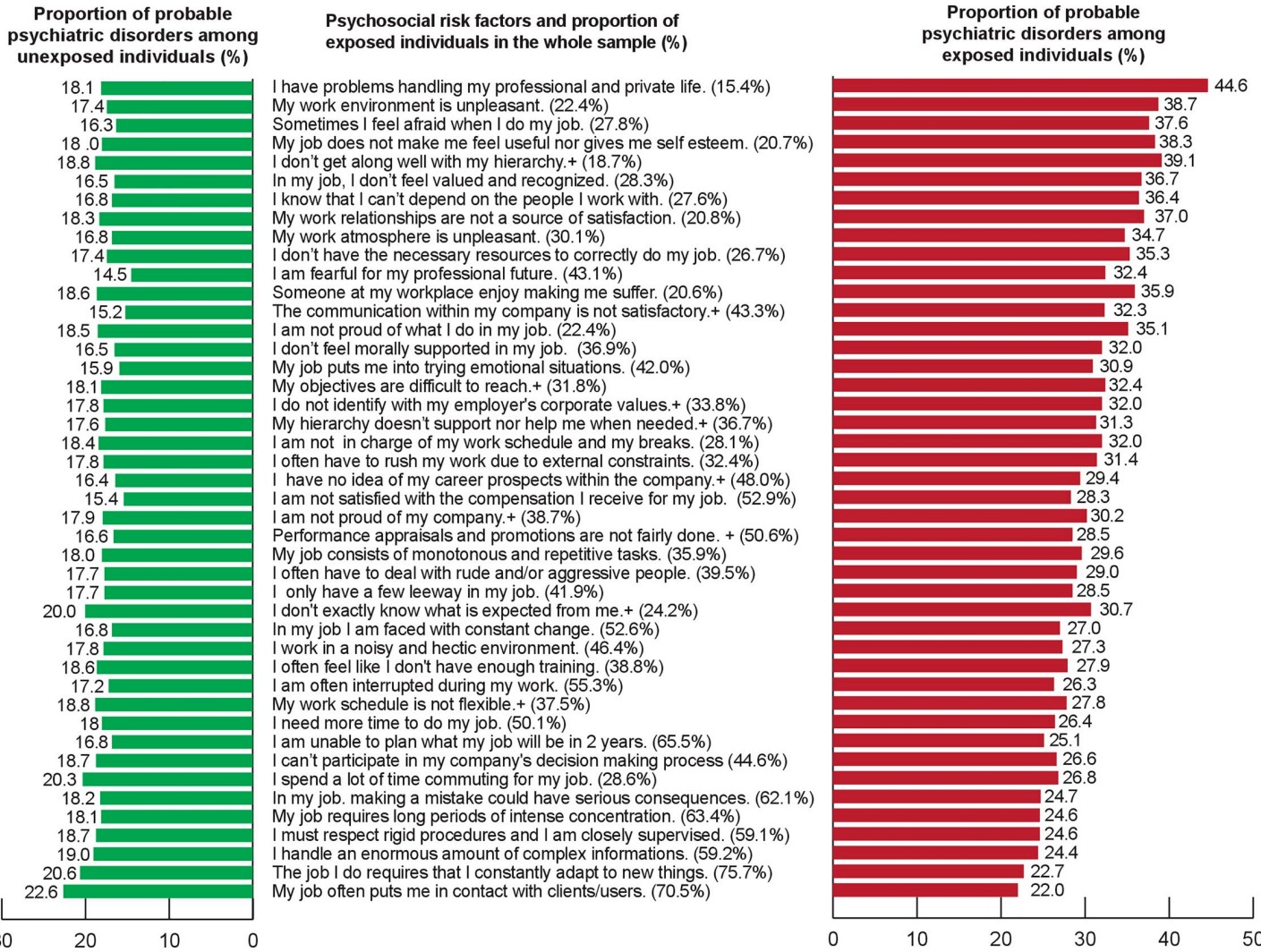

**Fig 2. Ranking of the absolute difference between the proportion of probable psychiatric disorders among exposed and non-exposed individuals.** Proportions of probable psychiatric disorders are given for each group (exposed and non-exposed) as well as the proportion of participants exposed in the whole sample. For example, 15.4% of individuals reported having problems handling professional and private life. Among them, 44.6% may be cases as compared with 18.1% among non-exposed individuals. Hence, this psychosocial risk factor is ranked first place with the largest absolute difference (44.6–18.1).

people reporting not feeling useful were probable psychiatric disorder and only 18% among non-exposed individuals. Overall, 39.1% were probable psychiatric disorder among people reporting not getting along with their hierarchy, and only 18.8% among non-exposed individuals.

## Association of work-related PSRFs and probable psychiatric disorder (GHQ-28 score ≥24)

The fast-backward procedure identified 11 PSRFs to be included in the model (Table 2) and the results of the adjusted logistic regression of the 11 PSRFs on the outcome GHQ-28 score ≥24 are in Table 2 (S2 Table for the ORs and 95% CIs for the adjustment variables). Ten PSRFs had ORs > 1, with 95% CIs strictly over 1. Three PSRFs had ORs > 1.5: "problems handling professional and private responsibilities" (OR = 1.97 [1.52; 2.54]), "can't depend on

**Table 2. Logistic regression of psychosocial risk factors on the outcome GHQ-28 score $\geq$24 for employees.**

| Psychosocial risk factors | OR | 95% CI | | P value | Proportion of people exposed (n = 3200) |
|---|---|---|---|---|---|
| I have problems handling my professional and private responsibilities | 1.97 | 1.52 | 2.54 | <0.001 | 15.4 |
| I know that I can't depend on the people I work with | 1.63 | 1.29 | 2.06 | <0.001 | 27.6 |
| Sometimes I feel afraid when I do my job | 1.53 | 1.21 | 1.93 | <0.001 | 27.8 |
| I am fearful for my professional future | 1.44 | 1.15 | 1.78 | 0.001 | 43.1 |
| My job puts me into trying emotional situations | 1.43 | 1.13 | 1.79 | 0.002 | 42.0 |
| I am not satisfied with the compensation I receive for my job | 1.42 | 1.15 | 1.77 | 0.001 | 52.9 |
| The communication and information exchange process within my company are not satisfactory | 1.39 | 1.11 | 1.75 | 0.004 | 43.3 |
| My job requires long periods of intense concentration | 1.35 | 1.08 | 1.70 | 0.009 | 63.4 |
| My job doesn't make me feel useful and nor gives me self-esteem | 1.32 | 1.03 | 1.69 | 0.029 | 20.7 |
| My job consists of monotonous and repetitive tasks | 1.29 | 1.04 | 1.60 | 0.021 | 35.9 |
| My work environment is unpleasant | 1.26 | 0.99 | 1.61 | 0.061 | 30.1 |

OR = odds ratio; 95% CI = 95% confidence interval

These PSRFs were previously selected by a fast-backward procedure. Odds ratios of the adjustment variables are in S2 Table.

colleagues" (OR = 1.63 [1.29; 2.06]) and "sometimes feeling afraid when doing the job" (OR = 1.53 [1.21; 1.93]). The other PSRFs were "being fearful of the professional future" (OR = 1.44 [1.15; 1.78]), "not satisfied with the job compensation" (OR = 1.42 [1.15, 1.77]), "placed in trying emotional situations during work" (OR = 1.43 [1.13; 1.79]), "unsatisfied with the communication and information exchange within the company" (OR = 1.39 [1.11; 1.75]), "a job that requires long periods of intense concentration" (OR = 1.35 [1.08; 1.70]), "a job consisting of monotonous and repetitive tasks" (OR = 1.29 [1.04; 1.60]) and finally "a job that does not make one feel useful or give self-esteem" (OR = 1.32 [1.03;1.69]).

For the interactions between sex and the 11 PSRFs, none was significant (S3 Table). For the interactions between the 11 PSRF themselves, 9 were significant (p<0.05) but none remain significant after Holm-Bonferroni correction (S4 Table).

## Discussion

This study allowed us to estimate the prevalence of probable psychiatric disorder (GHQ-28 score exceeding the threshold 24) in the working population in France: 22.2% [95% CI 20.6; 24.0]. This prevalence was congruent with 17.2% of participants having taken psychotropic medication during the 12 last months (antidepressants or anxiolytics) (Table 1). As a comparison, in a meta-analyses of epidemiological data between 1980 and 2013, the prevalence of mental disorders was estimated at 17.6% (95% CI 16.3–18.9) during the 12 months preceding the assessment [19]. This is the first French study made on a representative sample of the working population since previous study focused on specific population [7].

Our study found that probable psychiatric disorder were more numerous among women than men and those with a long commute time, working more than 50 hours a week and earning less than 15 000 euros per year (Table 1), which is consistent with the literature [18,19]. Atypical working time (weekends, staggered hours, night work) affects mental health [18]. In our study, the differences were significant, as for marital status (being alone), but not after correction by Holm-Bonferroni. We decided to use a stringent criterion to decide significance in order to avoid false discovery, and it could have also increased the β risk of ignoring a true difference.

In our study, the proportion of probable psychiatric disorder was higher among people exposed to all PSRFs as compared with those non-exposed, except for two PSRFs: being required to constantly adapt to new things and placed in contact with customers/clients/users. This finding could probably be explained by the unclear meaning of these sentences are ambiguous: it can be positive and negative to be placed in contact with customers for instance.

The imbalance of work and personal life embodied the strongest association with psychiatric disorder (OR = 1.97 [95% CI 1.52; 2.54]). These results are consistent with the literature from several developed countries about work–family conflicts [23–26]. Interpreting these results regarding the exposure, we found 44.6% of probable psychiatric disorder among exposed individuals but 18.1% among non-exposed, but 15.4% of the sample reported being exposed to this PSRF. Even if the exposure level of this factor is the lowest, its association with probable psychiatric disorder was the most important.

Social support in the workplace was associated with having a probable psychiatric disorder (OR = 1.63 [95% CI 1.29; 2.06], 27.6% of the sample being exposed) as well as communication (OR = 1.39 [1.11; 1.75], 43.3% exposed), which confirms the model of Theorell and Karasek [5]. The relevance of the job effort/job demand theory was confirmed by the strong significant association of the cognitive demand of the job (OR = 1.35 [1.08; 1.70], 63.4% exposed), performing monotonous and repetitive tasks (OR = 1.29 [1.04; 1.60], 35.9% exposed) and the importance of having a fair recognition of the work by financial (OR = 1.42 [1.15, 1.77], 52.9% exposed) or symbolic compensation (OR = 1.32 [1.03;1.69], 20.7% exposed) [4]. More than half of the sample was exposed to long periods of intense concentration and unsatisfactory job compensation, both PSRFs with a high OR (about 1.4), so they are critical PSRFs for mental health. For the latter, 28.3% of exposed individuals were probable psychiatric disorder, and only 15.4% were non-exposed.

Experiencing job insecurity (OR = 1.44 [95% CI 1.15;1.78], 43.1% of the sample being exposed) significantly affected the mental health outcome in our study, which is consistent with international findings in Europe and the United States [27–29]. Among people exposed, 32.4% were probable psychiatric disorder, and 14.5% were non-exposed.

Finally, we found an effect of emotional burden at work on mental health (OR = 1.43 [95% CI 1.13; 1.79], 42.0% exposed), which is consistent with other studies [30]. The latter is one of the most-investigated factors because of the increasing literature on burn-out. Since its first description in 1974, burn-out emerged as a specific syndrome related to work, with emotional exhaustion one of the three components [31]. However, it seems difficult to consider burn-out as a global indicator of the mental health status of workers. First, the relevance of distinguishing burn-out from depression by its attribution to work is debated. In fact, there is a strong overlap of its symptoms with those of depression [32]. Second, burn-out has no clear common definition and include a variety of symptoms. Finally, it is not recognized as a mental disorder: it does not appear in any classification (*Diagnostic and Statistical Manual of Mental Disorders*-5 or International Classification of Diseases, 11[th] revision). Therefore, we chose a global indicator of mental health, the GHQ-28, rather than a burn-out scale to investigate the association of PSRFs and mental health.

In this study we used a self-reported questionnaire about mental health (GHQ-28) to estimate the prevalence of psychiatric disorders in the sample. We choose a self-assessment instead of a hetero-assessment for feasibility reason (3200 participants). Knowing that only a standardized interview by a clinician can diagnose mental disorder, we use the wording "probable psychiatric disorder". Moreover, the positive predictive value of the GHQ-28 compared to the Composite International Diagnostic Interview is 48.8% to 66.4% [14]. For France, this PPV was 49.9% in France in 1997 for a threshold lower than our threshold of 24. Hence, for our study, the "true" prevalence of psychiatric disorder would be at least 11%. Considering that the

GHQ-28 measures symptoms of depression and anxiety (in fact its four dimensions are somatic symptoms, anxiety and insomnia, social dysfunction and severe depression), this "true" prevalence of our study is consistent with the proportion of 9.9% for depression and anxiety reported by the Organisation for Economic Co-operation and Development [1].

Most of the studies investigating the association of PSRFs and mental disorders are cross sectional. As in our study, they allow for identifying correlations between work-related factors and mental disorders. However, the determination of causality would require a longitudinal study, for instance, with a large cohort, to determine whether the exposure really preceded the mental disorder [18]. This investigation is particularly important for mental disorders such as depression in which negative cognitive bias and dysfunctional thoughts could bias the perception of the environment and thus the subjective restitution of work conditions.

## Conclusion

Our study estimated that one in five French workers have a probable psychiatric disorder. Moreover, we identified 10 PSRFs associated with having a probable psychiatric disorder. Among them, work–family conflict was the most important in terms of the intensity of the association, but had the lowest exposure rate in the sample. Lack of support from the company, job insecurity, and emotional burden of the job were also associated with poor mental health outcome, as was unfair job compensation or high cognitive demanding tasks. These results could offer useful suggestions for decision makers to manage and prevent mental disorders in the workplace. However, prospective studies are required to explore the causal effect of the psychosocial factors identified as strongly associated with probable psychiatric disorder and the efficacy of preventive interventions on these PSRFs.

## Supporting information

**S1 Fig. Flow chart of participants in the study.**
(PDF)

**S1 Table. Psychosocial risk factors assessed in the study.**
(PDF)

**S2 Table. Raw database, weighting factors and weighted database.**
(PDF)

**S3 Table. Results of the multiple linear regression estimating the weight of the covariates on the GHQ-28 score for employees and self-employed individuals.**
(PDF)

**S4 Table. Interactions between the 11 PSRFs and between sex and the 11 PRSFs.**
(PDF)

**S1 Raw data.**
(XLSX)

## Acknowledgments

Scientific committee that provided advice: Patrick Légeron, MD; Nicolas Brosset, MD; William Dab, MD, PhD; Gilbert Saporta, PhD, Jean-Christophe Sciberras.

## Author Contributions

**Conceptualization:** Patrick Légeron, Gilbert Saporta, Mounia N. Hocine.

**Formal analysis:** Astrid M. Chevance, Oumou S. Daouda, Alexandre Salvador, Yannick Morvan, Gilbert Saporta, Mounia N. Hocine.

**Funding acquisition:** Raphaël Gaillard.

**Methodology:** Astrid M. Chevance, Gilbert Saporta, Mounia N. Hocine.

**Project administration:** Astrid M. Chevance, Raphaël Gaillard.

**Supervision:** Raphaël Gaillard.

**Validation:** Astrid M. Chevance.

**Visualization:** Astrid M. Chevance.

**Writing – original draft:** Astrid M. Chevance.

**Writing – review & editing:** Astrid M. Chevance, Oumou S. Daouda, Alexandre Salvador, Patrick Légeron, Yannick Morvan, Gilbert Saporta, Mounia N. Hocine, Raphaël Gaillard.

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
