## [Decision Letter · Decision Letter 0]

23 Apr 2020

PONE-D-20-02569

Work-related psychosocial risk factors and prevalence of psychiatric disorders: A cross-sectional study in the French working population.

PLOS ONE

Dear Dr. Chevance,

Thank you for submitting your manuscript to PLOS ONE. After careful consideration, we feel that it has merit but does not fully meet PLOS ONE’s publication criteria as it currently stands. Therefore, we invite you to submit a revised version of the manuscript that addresses the points raised during the review process.

We would appreciate receiving your revised manuscript by Jun 07 2020 11:59PM. To enhance the reproducibility of your results, we recommend that if applicable you deposit your laboratory protocols in protocols.io, where a protocol can be assigned its own identifier (DOI) such that it can be cited independently in the future. For instructions see: http://journals.plos.org/plosone/s/submission-guidelines#loc-laboratory-protocols

We look forward to receiving your revised manuscript.

Kind regards,

Raoul Belzeaux, M.D.

Academic Editor

PLOS ONE

2. Please include your tables as part of your main manuscript and remove the individual files. Please note that supplementary tables (should remain/ be uploaded) as separate "supporting information" files

"“None”: Oumou Daouda, Mounia N. Hocine, Yannick Morvan and Alexandre Salvador report no conflict of interest

Astrid Chevance received a PhD grant from La Fondation pour la Recherche Médicale (FMD20170637634)

Raphaël Gaillard has been a member of a scientific board for Janssen, Lundbeck, Roche, SOBI and Takeda. He has been a consultant or speaker for Astra Zeneca, Boehringer-Ingelheim, Pierre Fabre, Lilly, LVMH, Lundbeck, MAPREG, Otsuka, Pileje, Sanofi, Servier and has received professional fees and has received funding for research from Servier. He is a founding member of Regstem. He is the president of the Fondation Pierre Deniker.

Gilbert Saporta is currently a consultant for Ipsos.

Patrick Légeron is a founding member of Stimulus."

Reviewers' comments:

Reviewer's Responses to Questions

**Comments to the Author**

1. Is the manuscript technically sound, and do the data support the conclusions?

Reviewer #1: Yes

Reviewer #2: Yes

2. Has the statistical analysis been performed appropriately and rigorously? 

Reviewer #1: Yes

Reviewer #2: Yes

3. Have the authors made all data underlying the findings in their manuscript fully available?

Reviewer #1: Yes

Reviewer #2: Yes

4. Is the manuscript presented in an intelligible fashion and written in standard English?

Reviewer #1: Yes

Reviewer #2: Yes

5. Review Comments to the Author

Reviewer #1: This is a very interesting manuscript, very well written, and bringing original evidence.

This is indeed the first epidemiological French study made on a representative sample of the working population. Representativeness was achieved by using quota sampling. The methodology is very sound and follows international standards. The authors conducted a STROBE-compliant cross-sectional study of a representative sample of the French workers, allowing to estimate the prevalence of probable psychiatric disorders in this population and among workers exposed to work-related psychosocial risk factors.

I have few comments:

- It is surprising to include individuals aged 18 to 80 years, in a study on active workers in a country where the age of retirement is around 60 to 62 years old. Moreover, the study excluded retired people. Nearly 5% of the study sample is aged more than 60. Can you clarify this point please ?

- In this study, 42.6% of participants had a work duration of more than 39h a week, while the legal work duration is 35h a week. Who's is concerned by this ? Workers, independent, managers ?

- One major result is that the imbalance of work and personal life embodied the most strongest association with psychiatric disorder. This is a clearly very relevant result even from clinical point of view, as it is frequently observed in workers suffering from burnout.

Reviewer #2: This is a very interesting study about the estimation of the prevalence of probable psychiatric disorders in the working population, and the assessment of the proportion of people presenting a probable psychiatric disorder among people exposed to work-related psychosocial risk factors. A strength of the study is that it is conducted in a large sample representative of the French population.

Only few points should be clarified.

First, authors should define patients exposed vs unexposed regarding the psychosocial risk factors (PSRF). I think that a patient who is exposed to one PSRF given is a patient with a score of 1 regarding this PSRF in the list of 44 questions. But, authors should write it clearly.

Second, I do not understand well this sentence: “These two PSRFs were the most reported, with 75.7% and 70.5% of the sample reporting having a job that required constantly adapting to new things and 70.5% reporting often being in contact with customers/clients/users.” This sentence should be re written.

Except these two points, this manuscript is very clear with a consistent statistical analysis.

Moreover, results are very interesting.

6. PLOS authors have the option to publish the peer review history of their article (what does this mean?). If published, this will include your full peer review and any attached files.

Reviewer #1: Yes: Wissam El-Hage

Reviewer #2: No

---

## [Author Response · Author response to Decision Letter 0]

1 May 2020

Responses to the reviewers:

We thank all reviewers for their careful reading. Their comments greatly helped us improve our work. Please find in blue our comments and in green the modification of the manuscript. 

Reviewer #1: 

This is a very interesting manuscript, very well written, and bringing original evidence.

This is indeed the first epidemiological French study made on a representative sample of the working population. Representativeness was achieved by using quota sampling. The methodology is very sound and follows international standards. The authors conducted a STROBE-compliant cross-sectional study of a representative sample of the French workers, allowing to estimate the prevalence of probable psychiatric disorders in this population and among workers exposed to work-related psychosocial risk factors.

We thank Reviewer 1 for the positive comments and encouragements. 

I have few comments:

- It is surprising to include individuals aged 18 to 80 years, in a study on active workers in a country where the age of retirement is around 60 to 62 years old. Moreover, the study excluded retired people. Nearly 5% of the study sample is aged more than 60. Can you clarify this point please ?

The eligibility criteria were: 

- Individuals aged 18 to 80 years AND

- Having a job (e.g. any kind of job, including part-time/self-employment)

Exclusion criteria: 

- Students

- Unemployed individuals

- Housewives/husband

- Retired people

A report of the French National Institute for Statistical and Economical Studies from 2018, showed that 5% of people aged [65-74] have an occupation (mainly men, with a good health condition, high education and living in the Parisian area). Top managers but also self-employed and farmers are overrepresented. Four profiles are described: employees with low education with part-time job, people with high education living in big cities, shopkeepers and farmers. (https://www.insee.fr/fr/statistiques/3646000?sommaire=3646226). 

To collect data on the senior workers, we included people over the legal retirement age, who declared an occupation. They represented 4.5% of our whole sample (see supplementary table 2 page 3) which is consistent with the national statistics.

- In this study, 42.6% of participants had a work duration of more than 39h a week, while the legal work duration is 35h a week. Who's is concerned by this ? Workers, independent, managers ?

We performed supplementary statistics on this population of workers who reported working more than 39h per week in the table hereunder. Our results are consistent with the report of the French National Institute for Statistical and Economical Studies from 2017 with a mean work duration of 50.5 hours a week for self-employed and 39.1 hours a week for employees. (Please see figure 5 in https://www.insee.fr/fr/statistiques/3676634?sommaire=3696937)

 Work duration ≥ 39 hours / week (n=1423) Work duration < 39 hours / week (n=1777)

Sex 

 Male 62.9 42.9

 Female 37.1 57.1

Age, years 42.0 (10.7) 40.9 (11.4)

Number of children 

 None 50.1 54.5

 1 23.4 20.5

 2 21.2 18.9

 3 4.6 4.9

 > 4 0.6 1.2

Highest educational degree 

 baccalaureate degree 18.2 23.6

 Baccalaureate degree 19.2 23.5

 Baccalaureate degree +2 years 24.6 23.9

 Baccalaureate degree ≥ 3 years 38.0 28.9

Annual household income (euros) 

 < 15 000 7.8 13.7

 15–24 000 17.4 22.8

 24–36 000 25.1 27.0

 > 36 000 37.7 23.0

 No information 11.9 13.5

Occupational status 

 Independent worker 16.2 7.4

 Employees 83.8 92.6

Size of company (no. of employees) 

 < 10 21.6 22.3

 10–49 20.0 24.5

 50–99 9.7 11.2

 100–499 19.3 21.3

 > 499 17.1 16.0

 No information 12.3 4.7

Activity sector 

 Industry 11.1 14.0

 Building 15.2 8.3

 Trading 12.1 11.7

 Transport 5.5 5.6

 Insurance and real estate 3.6 3.2

 Education, health and social work 19.5 26.4

 Other services 33.1 30.9

Duration of job 

 < 6 months 3.7 7.0

 6 months–5 years 33.2 33.9

 6–10 years 21.6 21.8

 > 10 years 41.5 37.3

Commuting duration 

 < 30 min 55.6 61.6

 30 min–1hr 21.4 22.2

 > 1hr 23.0 16.2

- One major result is that the imbalance of work and personal life embodied the most strongest association with psychiatric disorder. This is a clearly very relevant result even from clinical point of view, as it is frequently observed in workers suffering from burnout.

We agree with the reviewer that the imbalance of work and personal life is a major result of the study that echoed clinical practice. In this study we choose to focus on mental disorders and not on burn-out which is not part of the current mental disorder classification (neither the DSM-5 nor the ICM-10), as mentioned in the discussion. 

Reviewer #2: 

This is a very interesting study about the estimation of the prevalence of probable psychiatric disorders in the working population, and the assessment of the proportion of people presenting a probable psychiatric disorder among people exposed to work-related psychosocial risk factors. A strength of the study is that it is conducted in a large sample representative of the French population. Only few points should be clarified.

We thank the reviewer for acknowledging our work with positive comments.

First, authors should define patients exposed vs unexposed regarding the psychosocial risk factors (PSRF). I think that a patient who is exposed to one PSRF given is a patient with a score of 1 regarding this PSRF in the list of 44 questions. But, authors should write it clearly.

We add a sentence in the method section (paragraph 2) Measurement of psychosocial risk factors PSRF): 

“Therefore, participants with a score of 1 to a given PSRF are called “participants exposed” to this PSRF in the results section.”

Second, I do not understand well this sentence: “These two PSRFs were the most reported, with 75.7% and 70.5% of the sample reporting having a job that required constantly adapting to new things and 70.5% reporting often being in contact with customers/clients/users.” This sentence should be re written.

We corrected the sentence as followed (results section, “exposure to work-related PSRFs): 

“These two PSRFs were the most reported, with 75.7% of the sample reporting having a job that required constantly adapting to new things and 70.5% reporting often being in contact with customers/clients/users. “

Except these two points, this manuscript is very clear with a consistent statistical analysis.

Moreover, results are very interesting.

Response to the editor: 

- We formated the manuscript according to Plos One standard. 

- We decided to make the the data set available (open access) as supporting information.

- We add the specific sentence to the conflict of interest statement.

---

## [Editor Report · Decision Letter 1]

6 May 2020

Work-related psychosocial risk factors and psychiatric disorders: A cross-sectional study in the French working population.

PONE-D-20-02569R1

Dear Dr. Chevance,

We are pleased to inform you that your manuscript has been judged scientifically suitable for publication and will be formally accepted for publication once it complies with all outstanding technical requirements.

With kind regards,

Raoul Belzeaux, M.D.

Academic Editor

PLOS ONE
---

## [Editor Report · Acceptance letter]

12 May 2020

PONE-D-20-02569R1 

Work-related psychosocial risk factors and psychiatric disorders: A cross-sectional study in the French working population. 

Dear Dr. Chevance:

I am pleased to inform you that your manuscript has been deemed suitable for publication in PLOS ONE. Congratulations! Your manuscript is now with our production department. 

With kind regards,

on behalf of

Dr. Raoul Belzeaux 

Academic Editor

PLOS ONE